# Objective pupillometry shows that perceptual styles covary with autistic-like personality traits

Chiara Tortelli[1], Marco Turi[2], David Charles Burr[3]*, Paola Binda[4]

[1]Department of Surgical Medical Molecular and Critical Area Pathology, University of Pisa, Pisa, Italy; [2]Fondazione Stella Maris Mediterraneo, Chiaromonte, Italy; [3]Department of Neuroscience Psychology Pharmacology and Child Health, University of Firenze, Firenze, Italy; [4]Department of Translational Research on New Technologies in Medicine and Surgery, University of Pisa, Pisa, Italy

**Abstract** We measured the modulation of pupil size (in constant lighting) elicited by observing transparent surfaces of black and white moving dots, perceived as a cylinder rotating about its vertical axis. The direction of rotation was swapped periodically by flipping stereo-depth of the two surfaces. Pupil size modulated in synchrony with the changes in front-surface color (dilating when black). The magnitude of pupillary modulation was larger for human participants with higher Autism-Spectrum Quotient (AQ), consistent with a local perceptual style, with attention focused on the front surface. The modulation with surface color, and its correlation with AQ, was equally strong when participants passively viewed the stimulus. No other indicator, including involuntary pursuit eye movements, covaried with AQ. These results reinforce our previous report with a similar bistable stimulus (Turi, Burr, & Binda, 2018), and go on to show that bistable illusory motion is not necessary for the effect, or its dependence on AQ.

*For correspondence:
dave@in.cnr.it

## Introduction

Many images can be perceived at either a global or local level, as in the classic example of the forest and the trees. Typical development brings about a gradual shift from local to global focus (see *Goodenough, 1976*; *Kimchi, 2015*; *Wagemans et al., 2012* for reviews). However, individuals differ in their tendency to focus locally or globally, and there is growing evidence that individuals with autism show biases toward local processing (for meta-analysis, see *Van der Hallen et al., 2015*). Autistic individuals outperform neurotypicals in tasks where fine-grained visual features must be abstracted from their global context, such as the embedded figure task (*Shah and Frith, 1983*) or the Navon figures (*Plaisted et al., 1999*). However, it has been pointed out that many of these are sensitive to task instructions, and may also be affected by cognitive strategy (see *Baisa et al., 2019*; *Horlin et al., 2016* for reviews). Therefore, there is a pressing need for objective and quantitative indices of the local-global preference.

In our previous study (*Turi et al., 2018*), we introduced a pupillometry-based index that revealed a strong relationship between local-global processing styles and autistic-like traits, assessed in neurotypical adults by the Autism-Spectrum Quotient (AQ) questionnaire (*Baron-Cohen et al., 2001*). Observers viewed a classic bistable motion illusion (*Andersen and Bradley, 1998*; *Treue et al., 1991*), comprising transparent fields of black and white dots moving in opposite directions, giving the impression of a cylinder rotating about its vertical axis, with the direction of rotation and the color of the front surface changing periodically. Pupil size modulated in synchrony with perceptual reports, dilating when the front surface was black, constricting when white. Importantly, the magnitude of pupillary modulation varied with AQ, stronger for participants with higher AQ scores,

consistent with attention being focused on the front surface (local style). Participants with lower AQ scores showed smaller or no pupil modulations, consistent with attention being distributed over the front and rear surface of the cylinder at all times (global style).

That study introduced a simple pupillometric measure that could potentially provide a quantitative and objective index of local-global preference in both neurotypical and autistic individuals. However, the illusory and bistable nature of the stimulus could be a limitation of the technique. Firstly, the neural dynamics underlying these phenomena are complex and only partially understood (*Brascamp et al., 2018*), and it is unclear whether any of these perceptual or decisional factors drive the pupil modulations. In addition, susceptibility to visual illusions also correlates with autistic traits (*Chouinard et al., 2016*; *Happé, 1996*; *Mitchell et al., 2010*; *Pellicano and Burr, 2012*), making it difficult to judge whether the differences in pupil modulation reflect differences in susceptibility to illusions rather than differences in perceptual style. Finally, although the pupillometric measurements are objective, their analyses relied on the subjective reports of perceptual alternations, which may be unreliable and difficult to obtain from some participants, particularly young children and clinical groups.

To address these concerns, we adapted the technique so it does not rely on illusory reversals, by disambiguating the 3D rotation of the cylinder with binocular disparity cues. Observers viewed the display through a stereoscope, and stereo-depth was periodically swapped to bring either the black or white dots to the foreground, which also changed the direction of rotation of the cylinder. This modified stimulus allowed us to analyze changes in pupil size based on objective changes of stereo-depth, rather than perceptual reports. It also allowed us to eliminate completely the need for participants to report their perception, simply observing the alternating stimulus for a period of time.

We hypothesized that if the effect were driven by differences in perceptual style, the 3D layout of the stimulus should evoke a similar pupil modulation to that we found in bistable perception, and this should correlate with AQ scores in neurotypical adults, irrespectively of whether they were required to make perceptual reports. We also tested whether other voluntary and involuntary responses (including spontaneous slow eye movements) could provide alternative indices of perceptual style, correlated with AQ. Our results confirmed the previous report that pupillometry measures correlate well with AQ, revealing a strong association between perceptual styles and autistic traits. Spontaneous eye-drift tracked the direction of motion, but these were unrelated to AQ.

## Results

We used a modified version of the stimulus used in *Turi et al., 2018*, where clouds of white dots and black dots moved in opposite directions to generate the bistable illusion of a cylinder rotating in 3D about its vertical axis, with direction of rotation alternating between clockwise and counterclockwise. In the present study, the direction of rotation was unambiguous, defined by periodically changing stereo-cues, following similar dynamics to the perceptual switches observed in our previous study. The association between motion direction and color remained constant within each trial, but varied across trials; consequently, switches in stereo-depth over a trial implied changes in both color and motion direction of the foreground. Fifty-three participants reported the changes in rotation direction during 60 s trials, while we monitored pupil size and eye movements (*Figure 1*).

*Figure 2A* shows the average timecourse of pupil size synchronized to the stereo-depth swap and averaged over all participants, separately for phases with stereo-defined black or white foreground (blue and red, respectively). This figure has the same format as Figure 1B in *Turi et al., 2018*, but while *Turi et al., 2018* aligned the traces to the perceptual reports of a bistable stimulus, here they are aligned to physical changes in the stimulus, ignoring perceptual reports. We define zero as the end of the stereo-switch, which lasted 1.2 s. This procedural difference leads to differences in the overall shapes of the pupil timecourses, but the basic result is the same: pupil size modulates with the brightness of the foreground, dilating when black (blue traces of *Figure 2A*) and constricting when white (red traces). The black trace at the top of *Figure 2A* shows the difference in pupil size for the two directions of rotation, which begins to emerge after the depth swap is complete, and remains for several seconds.

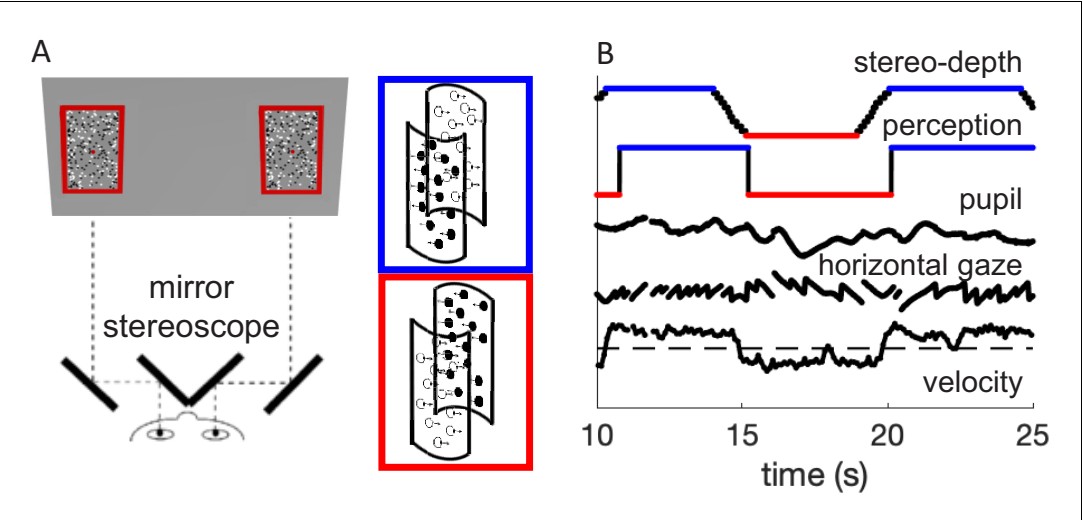

**Figure 1.** Methods. (**A**) Schematic diagram of the experimental setup, where participants viewed the screen through a mirror stereoscope while their left eye was tracked by a remote Eyelink camera. Note the small red fixation spot at the center of each display. Black and white dots moved in opposite directions, defining two overlapping surfaces that formed the front and rear of a rotating cylinder. Using stereopsis, we caused one of the surfaces to appear in front and the other behind the fixation plane, intermittently swapping the perceived rotation direction of the cylinder, as illustrated by the insets with colored outlines. (**B**) Example timecourses of stereo-depth (note the gradual change in disparity, lasting 1.2 s), perceptual reports, pupil size, horizontal gaze position (displaying OKN-like pattern), and velocity of the slow gaze shifts (fast transients filtered out).

The pupil-size modulations are consistent with attention being focused on the front surface, as a relative pupil dilation or constriction is expected from attending to black or white, respectively (*Binda et al., 2013*; *Mathôt et al., 2013*). It is important to reiterate that here, as in *Turi et al., 2018*, pupil-size modulations cannot be explained by changes in luminance, as it was constant during the whole experiment, nor by changes in the luminance distribution over space, as the white and black dots formed two overlapping transparent surfaces.

As in our previous study, there were large interindividual differences in the amplitude of pupil-size modulations, and again a large portion of the variance could be accounted for by the participants' AQ score. *Figure 2B* plots the pupil modulation (difference in size for the two directions of motion) against AQ of each participant. The correlation is positive, strong, and highly significant ($r = 0.45$, $p<0.001$, base-10 logarithm of the Bayes Factor [lgBF] = 1.49), showing that the magnitude of the effect increases with AQ. This is consistent with the hypothesis that the magnitude of pupil-size modulation depends on the tendency to focus attention locally on the front surface (hence driven by surface color) or globally across both surfaces, and that these two strategies correspond well to the local and global perceptual styles associated with high and low autistic traits (*Grinter et al., 2009*; *Happé and Frith, 2006*).

That the association between pupil-size modulation and AQ holds in spite of the differences of the two studies has important implications. Firstly, it implies that the illusory bistable nature of the stimulus in *Turi et al., 2018* is not the primary force driving the modulations in pupil size (although it might play a secondary role, accounting for the much higher correlation coefficient observed in our previous study). Secondly, it suggests that modulation of pupil size is not yoked to processes of perceptual decision and report, given that it can be driven by information about the physical stimulus only (see also below).

How specific is pupil-size modulation in revealing these AQ-linked interindividual differences? As in our previous study, we verified that the overall pupil responsivity could not account for this effect, as the mean pupil response (the size of the overall pupil dilation in the 1–2 s window following the depth swap) was reliably uncorrelated to AQ scores ($r = -0.12$, $p=0.39$, lgBF = $-0.81$, *Figure 3—figure supplement 1*, left column). Moreover, in line with the previous findings, perceptual performance was uncorrelated with AQ ($r = 0.12$, $p=0.42$, lgBF = $-0.82$; here we measured performance as accuracy, by cross-correlating perceptual reports with stimulus stereo-depth timecourses and taking the peak of the function).

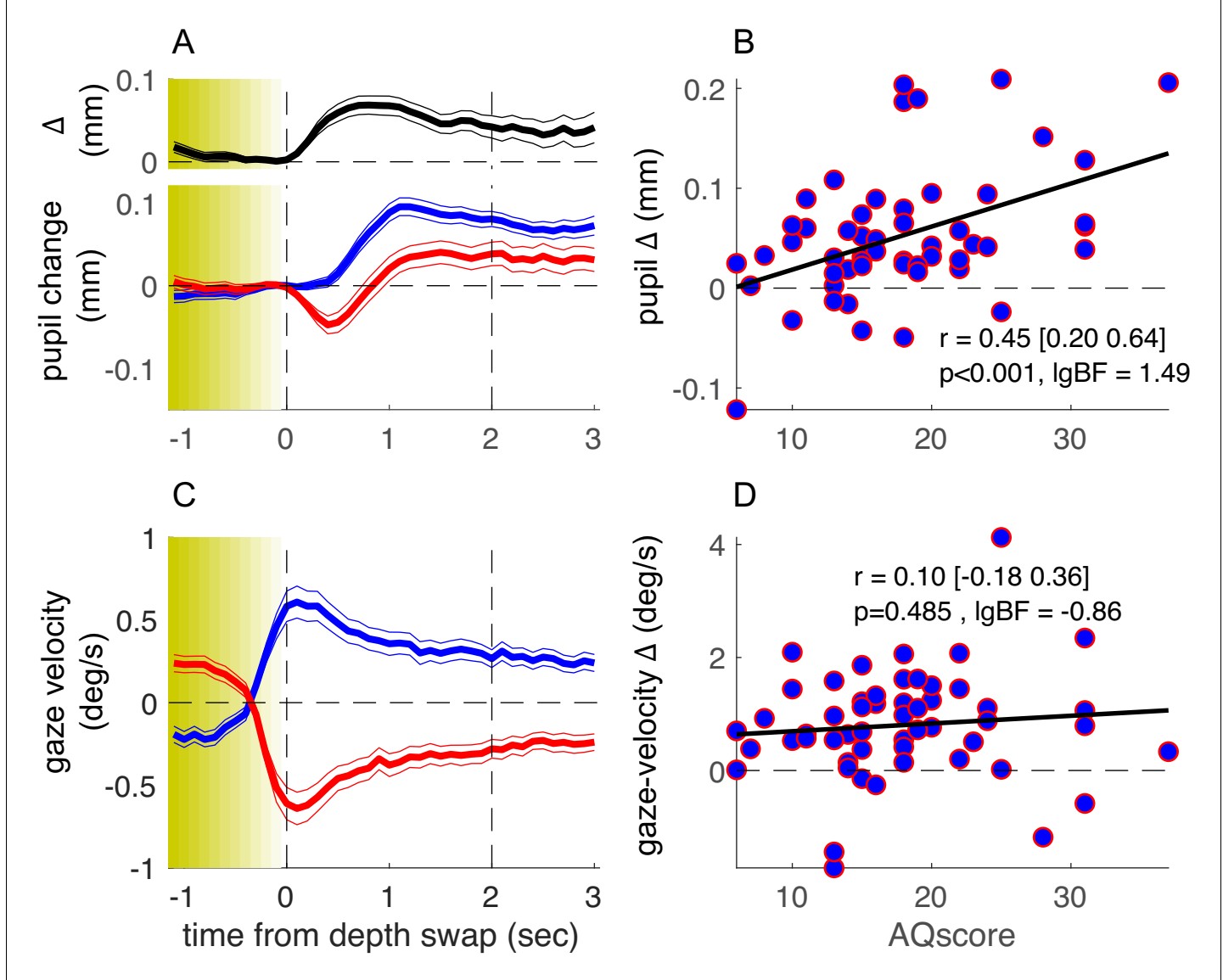

**Figure 2.** Results of the main experiment. (A) Pupil-size traces, synchronized to stereo-depth swaps (which occurred gradually over the yellow-shaded window) and averaged separately for times when the foreground was black (blue trace) or white (red trace). The top inset (black trace) shows the timecourse of the difference between the two traces. Thin lines show standard error across the 53 participants. (C) Same as A but showing gaze-velocity traces, computed after filtering out gazes shift faster than the maximum stimulus speed (3.9 deg/s) and after inverting the sign of velocity values for trials when the black dots moved leftward, so that positive gaze shifts always imply gaze shifts in the direction of the black dots. (B-D) Average pupil-size difference (B) or gaze-velocity difference (D), computed in the interval 0:2 s (marked by the vertical dashed lines in A and C) for each participant and plotted against their score on the Autism-Spectrum Quotient questionnaire. The thick black line shows the best fitting linear regression; text inset gives Pearson's correlation coefficient with 95% confidence interval and associated p-value and base-10 logarithm of the Bayes Factor (lgBF).

The online version of this article includes the following figure supplement(s) for figure 2:

**Figure supplement 1.** Results of the main experiment with report-based analysis.

We also monitored eye movements during the experiment, to see if they too may act as a predictor of perceptual style. Although participants were instructed to maintain fixation continuously on a salient mark throughout the experiment, most made periodic horizontal eye movements, with a slow phase in the direction of the cylinder rotation and a fast return phase (antagonistic OKN behavior, typical of transparent motion displays [*Niemann et al., 1994*]: see *Figure 1B* for example). We analyzed the slow phases of these gaze shifts and found them generally consistent with the direction of the foreground dots (*Figure 2C*). Although there was interindividual variability in this behavior (with

some tracking the rear dots), this was not correlated with AQ scores (*Figure 2D*), with Bayes factor indicating that the lack of correlation was significant (r = 0.10, p=0.485, lgBF = −0.86). Consistently, factoring out gaze-velocity differences did not attenuate the correlation between pupil-size modulations and AQ (partial correlation: r = 0.44, p<0.01).

Although behavioral performance did not correlate with AQ, it is possible that the pupil modulations are driven, at least in part, by the participants' keypress responses. We investigated this possibility with additional analyses and data. First, we re-analyzed our pupil-size data synchronized to the perceptual switches, rather than the stereo-depth swaps (*Figure 2—figure supplement 1*). The difference between pupil-size traces still correlated significantly with AQ (r = 0.39, p<0.01, lgBF = 0.72), indicating that our pupil-size modulation index is robust and relatively independent of the method used for parsing traces (*Figure 3—figure supplement 1*, compare the left and the middle columns).

More informatively, we measured a subsample of about half of our participants with a 'no-report' paradigm, where they simply viewed the stimulus for the same amount of time, without reporting their perception. The detailed form of the phase-locked timecourses of pupil size (*Figure 3A*, individual blue and red traces) differs from those in *Figure 2*, probably reflecting the lack of a pupil-dilation component linked to making and reporting perceptual decisions (*Einhäuser et al., 2008*). However, the timecourse of the pupil-size difference was very similar across experiments (compare the top black trace in *Figures 2A* and *3A*), and its magnitude was clearly correlated with AQ score (*Figure 3B*). Correlation coefficients for the pupil-size difference and the other parameters were similar to the main experiment, where participants made active reports (*Figure 3—figure supplement 1*, compare leftmost and rightmost columns). The correlation between pupil-size difference and AQ scores did not differ across experiments (Fisher test: F = 0.3, p=0.76).

These results support the hypothesis that there are multiple independent components of the pupil-size modulations related to observation of our complex stimulus. There is a transient dilation related to the key-press action (absent in the no-report experiment), best seen when time-locking pupil traces to the perceptual switches as in *Figure 2—figure supplement 1*: this component was not correlated with AQ (r = −0.15, p=0.29, lgBF = −0.72). There was also a transient constriction related to the perceptual change, isolated in the no-report experiment in *Figure 3*, which again did not correlate with AQ (r = −0.32, p=0.13, lgBF = −0.31). Finally, there was the relatively sustained pupil-size modulation, measured by the difference between traces for the two possible 3D layouts of the stimulus, and similarly observed in the two experiments. AQ scores were specifically correlated with this component (r = 0.38 or larger, p<0.05 and lgBF >0.6 in all cases), irrespective of whether perceptual reports were used for analysis or dispensed with altogether (see *Figure 3—figure supplement 1* for a summary of the AQ-correlation analyses).

Data from the no-report experiment also provide a lower-bound estimate of the test-retest reliability of our pupillometric and gaze-tracking indices. *Figure 4* shows the correlation between the main and no-report experiment (middle column) for all parameters. In addition, we estimated their internal consistency, by computing split-half reliabilities (left column). Although all parameters are significantly correlated across experiments, pupil-size difference has the highest test-retest reliability, in spite of having the lowest split-half reliability. Indeed the disattenuated correlation between experiments (right column of *Figure 4*), computed with the Spearman-Brown formula (*Spearman, 1910*), is particularly high (r = 0.93). This indicates that pupil-size modulations are noisy but consistent over time, suggesting that they are driven by stable individual differences. In contrast, the other indices, although more consistent within any experimental session, are far less consistent over time, possibly affected by the participants' time-varying state, or the task requirements.

## Discussion

In two experiments reported here, we built on our earlier demonstration (*Turi et al., 2018*) that pupillometry can be a useful tool to monitor perceptual style, which we show to be associated with autistic traits in neurotypical adults. We generalized this method to a simpler paradigm that does not rely on perceptual reports, and demonstrated its specificity relative to other indices of voluntary and spontaneous behavior.

Participants observed a 3D rotating cylinder whose direction of rotation and color of front and rear surfaces alternated periodically. Pupil size modulated with the alternations, constricting when

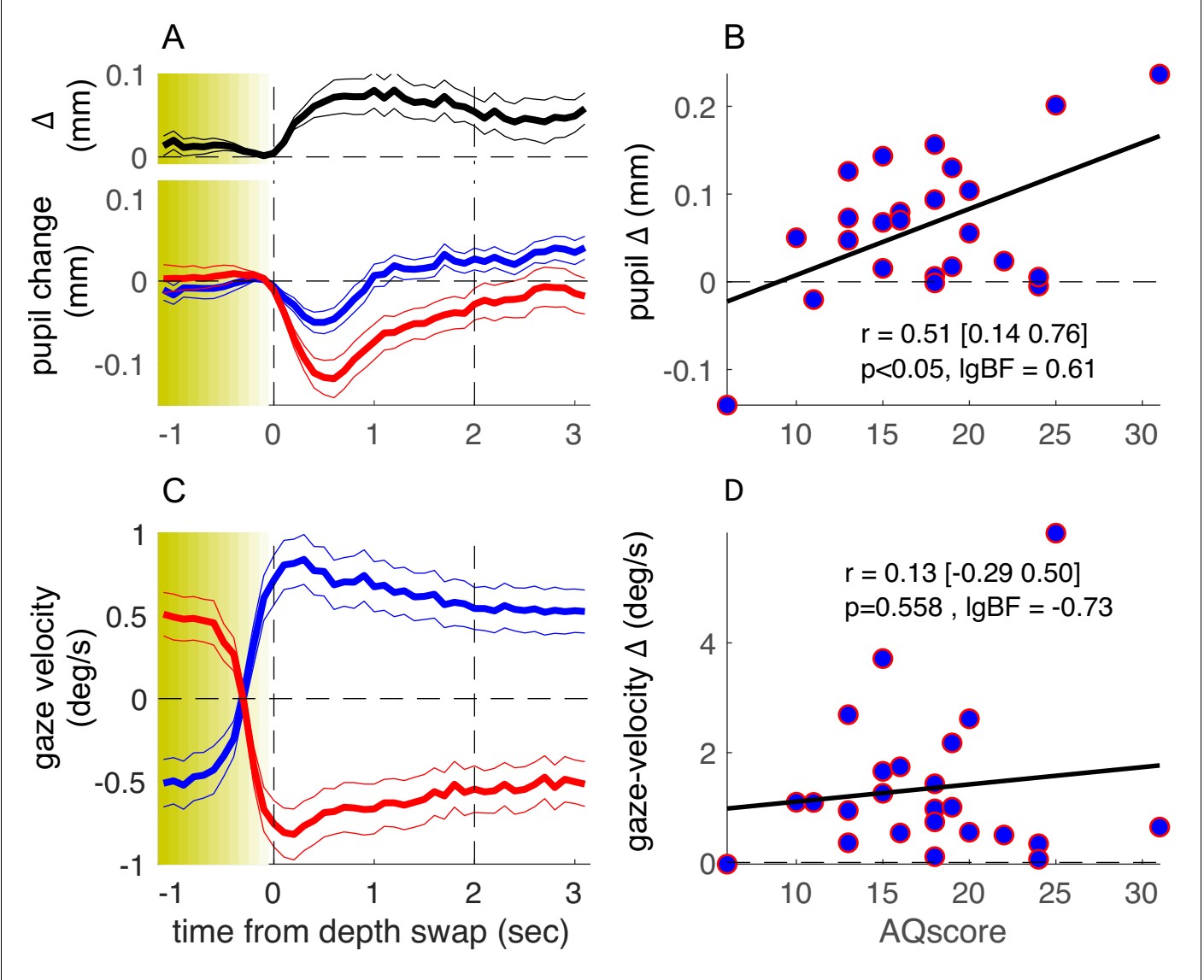

**Figure 3.** Results of the no-report experiment. Same as in *Figure 2* but showing results from a no-report version of the experiment, in which 24 of the original participants took part. lgBF: base-10 logarithm of the Bayes Factor .

The online version of this article includes the following figure supplement(s) for figure 3:

**Figure supplement 1.** Summary of the Autism-Spectrum Quotient (AQ) correlations.

the front surface was white, dilating when black. As in our previous work, pupil-size modulation was greater in high AQ individuals, suggesting that they attended more locally to the front surface, and less in individuals with low AQ, consistent with attention spread more globally over the whole cylinder. Whereas in the previous study the 3D layout of the stimulus was illusory and bistable and pupil-size modulation revealed by aligning pupil recordings with perceptual reports, here we disambiguated the 3D layout with stereo-depth, and aligned the pupil recordings to the physical changes. The correlation with AQ remained strong and significant, even when participants observed passively the stimuli without making any perceptual judgments. These results show that neither the complex neural dynamics underlying bistable perception (*Brascamp et al., 2018*) nor the act of reporting a perceptual decision is critical for inducing a modulation in pupil size. The reproducibility of pupil-size differences across our two experiments is testimony to its stability over time (which may be as high as r = 0.93 when the noise in each individual measurement is taken into account).

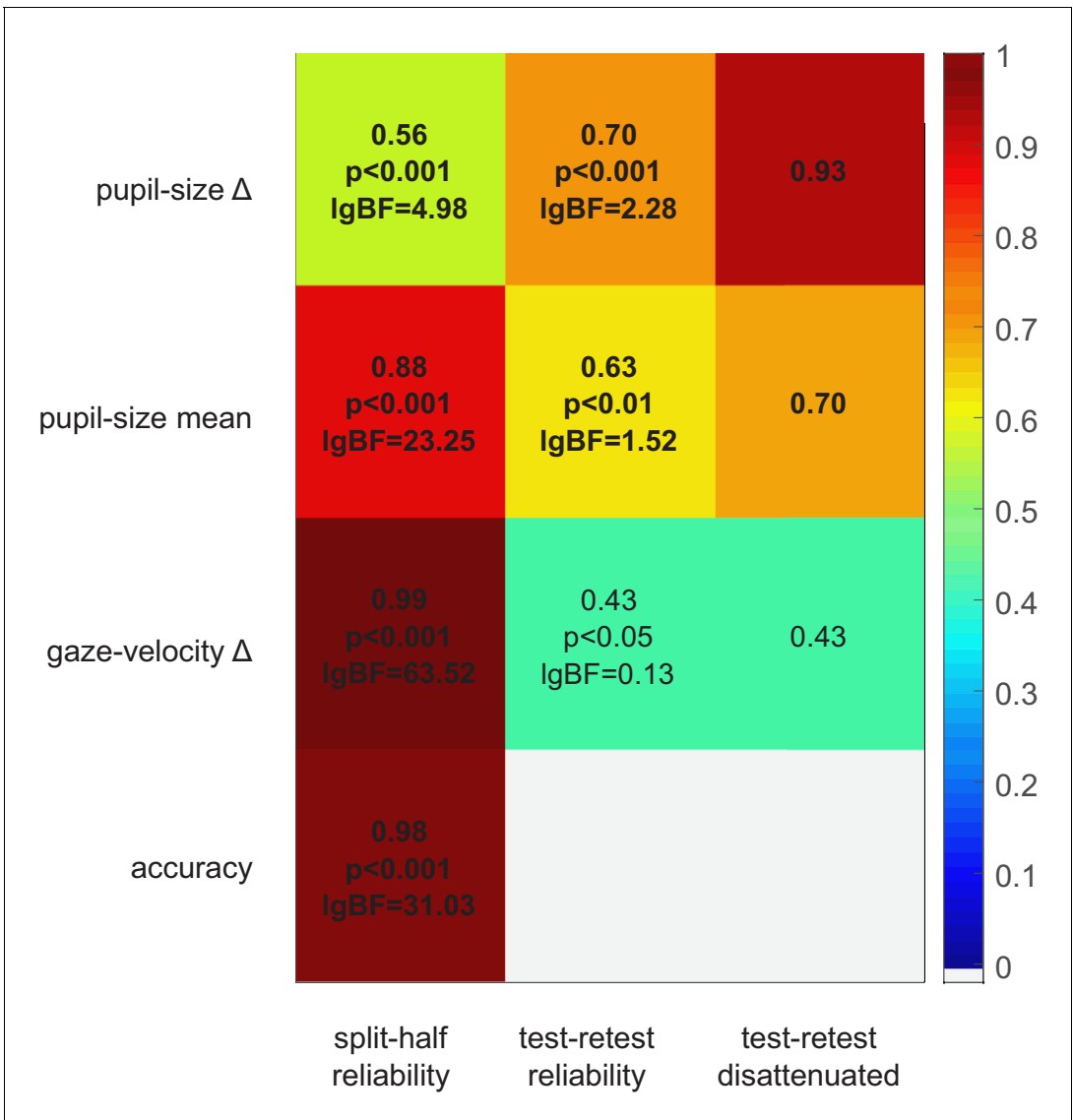

**Figure 4.** Split-half and test-retest reliability. Split-half reliability (computed as the Pearson's correlation coefficient of parameters computed from the odd and even trials, across participants and experiments), test-retest reliability (Pearson's correlation coefficient of parameters computed in the main and no-report experiment, across the 24 participants who took part in both), and disattenuated test-retest reliabilities (computed with the Spearman-Brown formula [*Spearman, 1910*] taking into account the split-half reliability of each parameter in each experiment).

By contrasting data collected with and without perceptual reports, and across different methods of analysis, we were able to highlight at least three components contributing to the pupil-size modulations. One is a transient pupil dilation, time-locked to the reporting of a perceptual decision, and previously associated with increased arousal and norepinephrine release (*Einhäuser et al., 2008*). Another is a transient pupil constriction, time-locked to the occurrence of a stimulus change that may be interpreted as an 'onset' or 'grating response' (*Barbur et al., 1992*), associated with visual cortical activity (*Sahraie et al., 2013*). Finally, there is a more sustained pupil-size difference depending on the 3D layout of the stimulus; this is compatible with the effect of attending to bright against dark (*Binda et al., 2013*; *Mathôt et al., 2013*), mediated by a cortical circuit (*Binda and Gamlin, 2017*). Only the third of these, the stimulus-dependent modulation of pupil size, correlated significantly with participants' AQ. Note that the pupil-size difference is not easily explained as a pupillary near response (the pupil constriction that accompanies near vision, *Marg and Morgan, 1949*). For this to influence the results, participants (especially those with higher AQ) would need to follow a particular color continuously, whether in front or behind. However, the gaze-velocity data shows that

most participants (irrespectively of their AQ) tended to track the horizontal motion of the front surface (*Figures 2D* and *3D*), suggesting there was no stimulus-driven change in vergence.

This OKN-like behavior (*Niemann et al., 1994*) could in principle have been informative of how participants distributed attention (*Mestre and Masson, 1997*). However, unlike pupil-size changes, gaze shifts are at least partially under voluntary control and were explicitly discouraged by asking participants to maintain fixation on a central mark. The tendency or ability to follow the instructions depends on a number of factors (e.g. motivation), which may be unrelated to AQ and might have masked any underlying correlation with AQ scores. On the other hand, the finding that gaze behavior is unrelated to AQ also indicates that differences in gaze behavior could not have generated differences in pupil-size modulations, as these were not necessarily stronger in participants who tended to track the foreground dots more systematically.

In summary, besides replicating (twice) the association between pupil-size modulations and autistic traits, the two experiments reported here confirm that pupillometry provides an index of local-global preference, which is reliable and stable over time, as may be expected for a parameter driven by genuine individual differences. In addition, data from the no-report experiment may be considered proof of principle that pupil measurements are informative even in a situation that requires minimal cooperation of the participants, which could pave the way for its implementation in a clinical sample with Autism Spectrum Disorder (ASD).

## Materials and methods

### Participants

Experimental procedures were approved by the regional ethics committee (*Comitato Etico Pediatrico Regionale—Azienda Ospedaliero-Universitaria Meyer—Firenze* [FI]) and are in accordance with the declaration of Helsinki; participants gave their written informed consent. Fifty-three neurotypical adults (41 women; age [mean ± SD]: 25.6 ± 3.3 years) volunteered to participate in the study. All reported normal or corrected-to-normal vision and had no diagnosed neurological condition. Sample size was set based on the previous study (we aimed for N = 50 and recruited three more to prevent data losses, which fortunately did not occur).

### AQ score

All neurotypical adult participants filled out the validated Italian version of the 50-item Autism-Spectrum Quotient questionnaire (*Baron-Cohen et al., 2001*; *Ruta et al., 2012*). Responses are made on a 4-point Likert scale: 'strongly agree,' 'slightly agree,' 'slightly disagree,' and 'strongly disagree.' Participants filled out the form on paper or online, just before or after completing the experiment. Items were scored as described in the original paper (*Baron-Cohen et al., 2001*): one when the participant's response was characteristic of ASD (slightly or strongly), 0 otherwise. Total scores ranged between 0 and 50, with higher scores indicating higher autistic traits.

There were no significant gender effects on either AQ scores (two-sample t-test: t(51) = 0.31, p=0.76, lgBF = −0.48) or on our pupil measurements (pupil difference in the main experiment: t(51) = 1.38, p=0.17, lgBF = −0.18). However, as the male gender was under-represented in our sample, these results should be considered with caution.

### Apparatus

The experimental setup was the same as in our previous study (*Turi et al., 2018*), with the exception that participants viewed a CRT (Cathode-ray tube) monitor (22-inchcolor monitor; 120 Hz, 800 × 600 pixels; Barco Calibrator) through a mirror stereoscope (ScreenScope SA200LCD – Stereo Aids). Stimuli were generated with the PsychoPhysics Toolbox routines (*Brainard, 1997*; *Pelli, 1997*) for MATLAB (MATLAB r2010a, The MathWorks).

Pupil diameter and position were monitored at 500 Hz with a video-based eyetracker (Eyelink 1000 SR Research). Pupil diameter was transformed from pixels to millimeters after calibrating the tracker with an artificial 4 mm pupil, positioned at the approximate location of the participants' left eye. Eye position recordings were linearized using a standard 9-point calibration acquired at the beginning of the experiment, before positioning the stereoscope (the calibration could not be performed through the restricted field of view of the stereoscope).

## Stimuli and procedure

Two 8 × 14 deg red outlines centered on a red fixation point (0.15 deg diameter) were presented against a gray background (12.4 cd/m2) and displaced by 7.2 deg horizontally so that, when viewed through the stereoscope, they fused into a single image. Within the rectangles, 150 white and black dots (each 0.30 deg diameter) moved in opposite directions with a sinusoidal velocity profile peaking at 3.9 deg/s, generating the perception of a cylinder rotating in 3D at 60 deg/s (*Figure 1A*). As in our previous study, the stimulus was displayed for 60 s long trials (30 per participant).

Unlike our previous study, we disambiguated the rotation of the cylinder with stereo-depth cues. We set the binocular disparity of the dot-cloud of one color (black or white) to 15′ arc, to place them in front of the fixation plane, with the opposite disparity −15′ arc for the other colored dots. When the left-moving dots were in front, the cylinder rotated clockwise, otherwise counterclockwise. The association between motion direction (left/right) and dot color (black/white) was varied pseudo-randomly across trials, but was constant throughout each trial. Stereo-depths were swapped every 5 s on average: 90% of intervals between swaps were drawn from a flat distribution with of mean 5 s and standard deviation (SD) 2 s, while the remaining 10% were assigned randomly between 0.5 and 60 s to reduce predictability. The dynamics of induced perceptual alternations were similar to that observed for the bistable perception of a rotating cylinder in the absence of stereo-depth cues (*Turi et al., 2018*). Disparities changed linearly over a 1.2 s interval, between ±15′ arc.

In the main experiment, 53 participants continuously tracked rotation direction by pressing one of three keys corresponding to clockwise, counterclockwise rotation, or mixed (in two participants, responses were not recorded due to a technical failure). Mixed percepts were very rare, on average 2.86 ± 0.63%, as expected given that the stereo-depth manipulation disambiguated perception. Participants were instructed to minimize blinks and maintain gaze on the fixation spot at all times, except during a 1 s intertrial pause, in which the cylinder disappeared.

A subsample (N = 24) was also tested in a 'no-report' version of the experiment, where they watched the same stimulus passively, without reporting rotation direction.

## Pupil and gaze tracking analyses

We analyzed data from four synchronous recordings: stimulus stereo-depth and key-press responses, pupil diameter, and x-gaze position.

Pupil diameter data were preprocessed with the following steps:

1. Identification and removal of gross artifacts: removal of time-points with unrealistically small or large pupil size (more than 1.5 mm from the median of the trial or <0.2 mm, corresponding to blinks or other signal losses).
2. Identification and removal of finer artifacts: identification of samples where pupil size varied at unrealistically high speeds (>2.5 mm/s, beyond the physiological range) and removal of the 20 ms epoch surrounding this disturbance.
3. Subtraction of mean and linear trends from each 60 s long trial: using the Matlab function 'detrend'

Preprocessing steps for x-gaze position data were:

1. Recoding: flipping the data for trials where black dots moved leftward, so that positive gaze-deviations imply deviations in the direction of the black moving dots.
2. Removal of high frequency noise: by convolving traces with a 20 ms box-car function.
3. Computation of velocity of the slow gaze shifts: by taking the derivative of gaze-deviations and removing all speeds higher than 3.9 deg/s (peak stimulus speed).

Pupil and slow-gaze velocity data, as well as continuous recordings of stimulus stereo-depth and perceptual reports, were down-sampled to 10 Hz, by taking the median of the retained time-points in nonoverlapping 100 ms windows. If no retained sample was present in a window, that window was set to 'NaN' (MATLAB code for 'not a number').

Traces were parsed into 10 s long segments, centered around a stereo-depth swap (when relative depth of white and black dots were swapped, using the end of a transition as reference) or a perceptual switch (when the subject changed reported perception). These were largely but not completely corresponding, since there was inevitably some variability in subjective reports. The resulting segments were labeled depending on whether the black or the white dots were in front (in terms of

stereo-depth or perceptually). Segments with more than 30% of missing pupil data were excluded (these were 5 ± 1% and 13 ± 2% for segments centered around stimulus and perceptual switches, respectively, leaving a total of 271 ± 5 and 270 ± 7 segments to be analyzed; for the no-report experiment, the percentage of discarded segments was 7 ± 2% for a total of 272 ± 7 analyzed segments). Segments of pupil data were further baseline corrected, by subtracting from each the median pupil in the 200 ms immediately preceding the swap. Finally, segments of pupil data, slow-gaze velocity, and perceptual reports were aggregated by computing the median across segments of the two types (with white or black dots in front). We quantified their separation by taking the difference over a 2 s long window following a stereo-depth swap or centred around a perceptual switch, i.e. with an offset of about 1 s, which corresponds well with the average latency of perceptual reports compared to stereo-depth swaps (see below). In addition, we quantified the pupillary response to stimulus or perceptual change by averaging both traces' second half of this window (1–2 s after the stereo-depth swap or 0–1 s after the perceptual switch, as in *Turi et al., 2018*).

To measure the accuracy of perceptual reports, we took the peak of the cross-correlation function between timecourses of perceptual reports and stereo-depth (computed after concatenating all trials and filling in missing values with linear interpolation). This value varied between 1 (reports in phase with stereo-depth) and −1 (reports out-of-phase), with 0 indicating chance behavior. On average, peak cross-correlation was 0.83 ± 0.02 (mean and standard error across participants). The latency of the peak gives an estimate of reaction times and it averaged 1.02 ± 0.06 s.

We estimated the internal consistency of our parameter estimates by split-half reliability. Each parameter was estimated twice per participant, on half the dataset (odd or even trials) and the two estimates correlated across participants (data from the main and the no-report experiments were entered as separate data-points).

Significance of the statistics was evaluated using both p-values and log-transformed Bayes Factors (*Wetzels and Wagenmakers, 2012*). The Bayes Factor is the ratio of the likelihood of the two models H1/H0, where H1 assumes a correlation between the two variables and H0 assumes no correlation. By convention, when the base 10 logarithm of the Bayes Factor (lgBF) >0.5 is considered substantial evidence in favor of H1, and lgBF < −0.5 substantial evidence in favor of H0.

## Acknowledgements

This project has received funding from the European Research Council (ERC) under the European Union's Horizon 2020 research and innovation program, Grant No 801715 (PUPILTRAITS) and Grant No 832813 (GenPercept), and from the Italian Ministry of University and Research under the PRIN2017 programme (grant MISMATCH) and FARE-2nd programme (grant SMILY, n. R182E5PNC7).

## Additional information

### Funding

| Funder | Grant reference number | Author |
| --- | --- | --- |
| H2020 European Research Council | 801715 | Paola Binda |
| H2020 European Research Council | 832813 | David Charles Burr |
| Ministry of Education, University and Research | MISMATCH | Paola Binda |
| Ministry of Education, University and Research | R182E5PNC7 | Paola Binda |

The funders had no role in study design, data collection and interpretation, or the decision to submit the work for publication.

## Author contributions
Chiara Tortelli, Data curation, Investigation, Writing - review and editing; Marco Turi, Conceptualization, Writing - review and editing; David Charles Burr, Conceptualization, Supervision, Funding acquisition, Writing - original draft, Writing - review and editing; Paola Binda, Conceptualization, Data curation, Software, Formal analysis, Supervision, Funding acquisition, Investigation, Writing - original draft, Writing - review and editing

## Author ORCIDs
Chiara Tortelli (iD) https://orcid.org/0000-0003-4921-5005
Marco Turi (iD) https://orcid.org/0000-0002-4495-0804
David Charles Burr (iD) https://orcid.org/0000-0003-1541-8832
Paola Binda (iD) http://orcid.org/0000-0002-7200-353X

## Ethics
Human subjects: Experimental procedures were approved by the regional ethics committee [Comitato Etico Pediatrico Regionale-Azienda Ospedaliero-Universitaria Meyer-Firenze (FI)] and are in accordance with the declaration of Helsinki. Participants gave their written informed consent.

## Decision letter and Author response
Decision letter https://doi.org/10.7554/eLife.67185.sa1
Author response https://doi.org/10.7554/eLife.67185.sa2

# Additional files

## Supplementary files
• Transparent reporting form

## Data availability
Experimental data have been uploaded to Zenodo at the following https://doi.org/10.5281/zenodo.4486576.

The following dataset was generated:

| Author(s) | Year | Dataset title | Dataset URL | Database and Identifier |
|---|---|---|---|---|
| Tortelli C, Turi M, Burr DC, Binda P | 2021 | Objective pupillometry shows that perceptual styles covary with autistic-like personality traits | https://doi.org/10.5281/zenodo.4486576 | Zenodo, 10.5281/zenodo.4486576 |

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
