## [Decision Letter]

**Acceptance summary:**

This Research Advance provides an important replication and advance on a biomarker associated with autistic-like personality traits. In an earlier study published in *eLife* by this research team, pupillometry was used as a proxy of perceptual style, with greater modulation of pupil size assumed to reflect a bias towards local over global processing. In a neurotypical population, this measure was found to correlate with scores on a standard psychometric assessment of autistic-like behaviors, the Autism-Spectrum Quotient (AQ) questionnaire. The present study provides a novel demonstration of this relationship while ruling out a number of alternative interpretations. Rather than rely on subjective reports associated with a bistable motion illusion, the current task controls the participants' perceptual experience by swapping the stereo-depth of two moving displays. Pupil size is primarily dictated by the luminance of the forward display. After confirming this relationship in the first experiment, the authors conduct a second experiment involving passive viewing, relying on the pupil size to indicate attentional focus. Again, they observe a positive correlation between pupil size modulation and AQ scores, as well as show that other eye movement features (e.g., velocity, mean pupil size) are not predictive of AQ scores. Taken together, the two experiments provide compelling and converging support for what appears to be a rather specific biomarker. The advance here rules out accounts of the relationship that might depend on task instructions, cognitive strategies, or task engagement. Moreover, given how this procedure can be quantified in a relatively simply manner, it is likely to be useful in future work involving individuals with autistic spectrum disorder.

**Decision letter after peer review:**

Congratulations, we are pleased to inform you that your article, "Objective pupillometry shows that perceptual styles covary with autistic-like personality traits", has been accepted for publication in *eLife*.

This Research Advance builds on a target article published in *eLife* by this research team (https://elifesciences.org/articles/32399). In the target article, pupillometry was used as a proxy of perceptual style, with greater modulation of pupil size assumed to reflect a bias towards local over global processing. In a neurotypical population, this measure was found to correlate with scores on a standard psychometric assessment of autistic-like behaviors, the Autism-Spectrum Quotient (AQ) questionnaire. The present study provides a novel demonstration of this relationship while ruling out a number of alternative interpretations. Rather than rely on subjective reports associated with a bistable motion illusion, the current task controls the participants' perceptual experience by swapping the stereo-depth of two moving displays. In this manner, pupil size is primarily dictated by the luminance of the forward display. After confirming this relationship in the first experiment, the authors conduct a second experiment involving passive viewing, relying on the pupil size to indicate attentional focus. Again, they observe a positive correlation between pupil size modulation and AQ scores, as well as show that other eye movement features (e.g., velocity, mean pupil size) are not predictive of AQ scores. Taken together, the two experiments provide compelling and converging support for what appears to be a rather specific biomarker. The advance here rules out accounts of the relationship that might depend on task instructions, cognitive strategies, or task engagement. Moreover, given how this procedure can be quantified in a relatively simply manner, it is likely to be useful in future work involving individuals with autistic spectrum disorder.

If the opportunity presents itself in the course of editing, I would recommend one point of clarification on the methods that I remain unclear on. Is the perceived direction of motion always consistent with the forward display? That is, is pupil size not only indicative of the forward display, but also of the direction of motion? I'm assuming so, but it would be nice to have this made explicit.

---

## [Author Response]

If the opportunity presents itself in the course of editing, I would recommend one point of clarification on the methods that I remain unclear on. Is the perceived direction of motion always consistent with the forward display? That is, is pupil size not only indicative of the forward display, but also of the direction of motion? I'm assuming so, but it would be nice to have this made explicit.

We addressed this comment: “The association between motion direction and color remained constant within each trial, but varied across trials; consequently, switches in stereo-depth over a trial implied changes in both color and motion direction of the foreground”.